# Development of Salt- and Gastric-Resistant Whey Protein Isolate Stabilized Emulsions in the Presence of Cinnamaldehyde and Application in Salad Dressing

**DOI:** 10.3390/foods10081868

**Published:** 2021-08-12

**Authors:** Huanhuan Cui, Qihang Liu, David Julian McClements, Bin Li, Shilin Liu, Yan Li

**Affiliations:** 1College of Food Science and Technology, Huazhong Agricultural University, Wuhan 430070, China; 17806275398@163.com (H.C.); 13203865687@163.com (Q.L.); libinfood@mail.hzau.edu.cn (B.L.); slliu2013@mail.hzau.edu.cn (S.L.); 2Department of Food Science, University of Massachusetts, Amherst, MA 01003, USA; mcclements@foodsci.umass.edu; 3Key Laboratory of Environment Correlative Dietology, Ministry of Education, Wuhan 430070, China; 4Functional Food Engineering &Technology Research Center of Hubei Province, Wuhan 430070, China

**Keywords:** whey protein, stability, ionic strength, gastric digestion, emulsions

## Abstract

Protein-stabilized emulsions tend to be susceptible to droplet aggregation in the presence of high ionic strengths or when exposed to acidic gastric conditions due to a reduction of the electrostatic repulsion between the protein-coated droplets. Previously, we found that incorporating cinnamaldehyde into the oil phase improved the resistance of whey protein isolate (WPI)-stabilized emulsions against aggregation induced by NaCl, KCl and CaCl_2_. In the current study, we aimed to establish the impact of cinnamaldehyde on the tolerance of WPI-stabilized emulsions to high salt levels during food processing and to gastric conditions. In the absence of cinnamaldehyde, the addition of high levels of monovalent ions (NaCl and KCl) to WPI-emulsions cause appreciable droplet aggregation, with the particle sizes increasing from 150 nm to 413 nm and 906 nm in the presence of NaCl and KCl, respectively. In contrast, in the presence of 30% cinnamaldehyde in the oil phase, the WPI-emulsions remained stable to aggregation and the particle size of emulsions kept within 200 nm over a wide range of salt concentrations (0–2000 mM). Divalent counter-ions promoted droplet aggregation at lower concentrations (≤20 mM) than monovalent ones, which was attributed to ion-binding and ion-bridging effects, but the salt stability of the WPI emulsions was still improved after cinnamaldehyde addition. The incorporation of cinnamaldehyde into the oil phase also improved the resistance of the WPI-coated oil droplets to aggregation in simulated gastric fluids (pH 3.1–3.3). This study provides a novel way of improving the resistance of whey-protein-stabilized emulsions to aggregation at high ionic strengths or under gastric conditions.

## 1. Introduction

Emulsion technology is widely utilized in the food and beverage industry to improve the desirability and palatability of foods [1,2,3]. It is also utilized to encapsulate, protect, and deliver functional food ingredients so as to enhance their handling, dispersibility, stability, bioavailability, and/or bioactivity [4,5]. Generally, emulsions consist of two phases, usually oil and water, with one of them being intimately dispersed in the other in the form of small droplets [5]. The formation and stabilization of the droplets in an emulsion-based food product is achieved by using food-grade emulsifiers that adsorb to the droplet surfaces and form a protective coating around them [6]. The nature of the emulsifier used for this purpose plays a critical role in determining the shelf life, physicochemical attributes, digestibility and gastrointestinal fate of food emulsions [7]. Many animals, plants, and microbial proteins are used as natural emulsifiers in foods because they are amphiphilic molecules that can adsorb to oil–water interfaces and form coatings that can generate strong steric and electrostatic repulsion under certain conditions, which inhibits droplet aggregation [6]. However, there are a number of limitations associated with using proteins as emulsifiers in foods [8]. In particular, protein-coated oil droplets are prone to aggregation under conditions where the attractive interactions between them are increased (e.g., when they are heated above their thermal denaturation temperature so as to expose nonpolar groups and strengthen hydrophobic attraction) or where the repulsive interactions between them are reduced (e.g., when the pH or salt concentration is altered in a way that weakens the electrostatic repulsion) [7,9]. A recent work by Doğan et al. confirmed the effect of salt (NaCl and CaCl_2_) on emulsions and also provided the effect mechanism of salt by affecting the interfacial rheological property of emulsions [10]. They demonstrated that the interfacial rheology results could predict the emulsion stability and CaCl2 salt increased the stability of emulsions stabilized by the selected emulsifiers (lecithin, mono-diglycerides, sodium steoryl-2-lactylate), as compared to NaCl salt. Moreover, Pickering emulsions have been known to have better stability against coalescence and aggregation. French et al. found that salt affected the particle–particle interaction, wettability of particle and also the particle–interface interaction. [11]. They concluded when considering the formation of Pickering emulsions, the influence of salt on the particle–particle interaction was less important than the influence of salt on both the particle wettability and the particle–interface interaction.

In this study, we focus on the impact of salts and gastric conditions on the stability of emulsions containing protein-coated oil droplets. Emulsions are exposed to different kinds of salts during food manufacture, storage, and preparation because they are often added to the aqueous phase of foods to alter their taste, stability, or properties [12]. Emulsions are also exposed to different kinds of salts within the gastrointestinal tract after foods are consumed because various kinds of mineral ions are naturally present within saliva, gastric fluids, and intestinal fluids [13]. In general, mineral ions vary in their dimensions and valences, which alters their ability to bind to proteins, form salt bridges between proteins, or screen electrostatic interactions [14]. It is often important to control the salt stability of emulsions in foods and the gastrointestinal tract.

The functional performance of food proteins can often be improved by forming physical complexes or covalent conjugates with other food molecules, including phytochemicals [15], other proteins [16], chitosan [17], maltodextrin [18], almond gum [19] and tannic acid [20]. Our previous research found that the incorporation of cinnamaldehyde (CA) into the oil phase of whey-protein-stabilized emulsions greatly improved their resistance to aggregation when exposed to high levels of monovalent (0–500 mM Na^+^ or K^+^) or divalent (0–30 mM Ca^2+^) counter-ions [21]. However, in some food products the mineral ion levels may exceed these values (e.g., concentration of Na^+^ reached 600 mM in sauce [10] and even 3000 mM in soy sauce products) or there may be other kinds of mineral ions present (e.g., Mg^2+^ and Fe^2+^). For this reason, we carried out additional experiments to establish the tolerance of whey-protein-stabilized emulsions to a wide range of ionic compositions.

Based on the knowledge gained from our previous work, we prepared whey protein isolate stabilized emulsions in the absence and presence of CA in the oil phase. The stability of the emulsions against ionic strength was first studied by characterizing particle size, zeta potential, microstructure and the droplet mobility. Then, the resistance of emulsions to aggregation under gastric digestion conditions was investigated. Finally, the emulsion-based salad dressing product was developed. The results of this study may be beneficial for the development of protein-stabilized emulsions suitable for application in high-salt foods.

## 2. Materials and Methods

### 2.1. Materials

Whey protein isolate (WPI, HilmarTM 9410) was purchased from Hilmar Ingredients (North Lander Avenue, CA, USA); it consisted of 93% protein in dry basis, 1.0% fat, 4.5% moisture, and 2.0% ash. Medium chain triglyceride (MCT) oil was obtained from Boxing Chemical Reagent Co., Ltd. (Wuhan, China). Cinnamaldehyde (CA, 95% purity) was purchased from Aladdin Reagent Co. (Shanghai, China). NaCl (99.5%), KCl (99.5%), CaCl_2_ (96%), MgCl_2_ (98%) and FeCl_2_ (99%) were obtained from Sinopharm Chemical Reagent Co., Ltd. (Shanghai, China). HCl (38.0%) and HNO3 (68.0%) solutions were also obtained from Sinopharm Chemical Reagent Co., Ltd. (Shanghai, China). Pepsin from porcine gastric mucosa (enzymatic activity of 3000 units per mg protein) was provided by the Aladdin Reagent Co. (Shanghai, China). Fluorescein isothiocyanate (FITC) and Nile red were purchased from Sigma-Aldrich Co. (St. Louis, MO, USA). Ultrapure water was used to prepare all solutions and emulsions, which was obtained from a Milli-Q-water purification system (Millipore, MA, USA). All other reagents used were of analytical grade.

### 2.2. Emulsion Preparation

Emulsions consisted of 5 wt% oil phase and 95 wt% aqueous phase. The oil phase consisted of MCT alone or a mixture of MCT and CA with a mass ratio of 7:3. The CA content was selected based on our previous work Chen et al. [21]. The aqueous phase (1 wt% WPI) was prepared by adding 1 g WPI powder (1 wt%) into 99 g phosphate buffer solution (PBS, 5 mM, pH 7.0) and then stirring at 4 °C overnight to ensure hydration and solubilization. A two-step method was used to prepare oil-in-water emulsions [21]. Firstly, coarse emulsions were prepared by blending oil (5 g oil) and water phases (95 g WPI solution) using a high-speed disperser (T18 digital ULTRA TURRAX, IKA Instruments Ltd., Staufen, Germany) operating at 12,000 rpm for 3 min. Secondly, the coarse emulsions were passed five times through a microfluidizer (Microfluidics M-110L, Microfluidics Corp., Newton, MA, USA) operating at 9000 psi to obtain fine emulsions, which was around 90 g. The emulsions were stored in an ice bath during these two steps. The emulsions were immediately adjusted to pH 7.0 after preparation. According to their oil phase compositions, the emulsions were named as M100C0 (100% MCT) and M70C30 (70 wt% MCT + 30 wt% CA).

### 2.3. Salt Stability

In order to study the resistance of the emulsions against ionic strength, monovalent ions (Na^+^, K^+^) and divalent ions (Ca^2+^, Mg^2+^, Fe^2+^) were selected as the ionic salts, and a wider range of salt concentrations was studied. Stock salt solutions were prepared by dissolving a weighed amount of powdered salt into double distilled water. The initial salt contents of the Na^+^, K^+^, Ca^2+^, Mg^2+^, and Fe^2+^ stock solutions were 5000, 5000, 500, 500, and 500 mM, respectively. The stock salt solutions were then diluted to the desired final concentrations. Then, 8 mL of emulsions and 8 mL of diluted salt solution were mixed together. The resulting mixtures were then continuously stirred for 15 min. The samples obtained contained the same oil content (2.5 wt%) but different mineral ion concentrations. The samples were then stored in a refrigerator at 4 °C for later use.

### 2.4. Particle Size and Zeta-Potential Measurements

The particle size and zeta-potential of the samples were measured 1 and 7 days after preparation. The zeta-potentials were measured using a Zetasizer Nano ZS instrument (Malvern Instruments, Worcestershire, UK). The samples were diluted 100-fold with buffer solution (5 mM, pH 7) to eliminate multiple light scattering effects. Refractive index values of 1.62 for CA and 1.45 for MCT were used. The refractive index of mixed oil phase in the M70C30 emulsions was 1.50, which was calculated as 1.62 × 0.3 + 1.45 × 0.7. After 30 s equilibrium, the samples were scanned three times and the averaged results were recorded. The particle size of the emulsions containing monovalent ions were measured using the dynamic light scattering module of the Zetasizer Nano ZS, since they were still in an appropriate size range. However, the particle size of the emulsions containing divalent ions was measured using static light scattering (MasterSizer 2000, Malvern Instruments, Worcestershire, UK) because the particles were much larger due to more extensive aggregation. The refractive indices of the oil and aqueous phases were again 1.50 and 1.33, respectively. The samples were diluted with distilled water until the opacity reached 4% to avoid multiple scattering effects. The particle size measurements were reported as the surface-weighted (d32) mean diameter.

### 2.5. Microstructure of Emulsions

The microstructure of the samples was visually observed using confocal laser scanning microscopy (CLSM, OLYMPUS FV3000, Olympus, Tokyo, Japan). The protein and oil phases were labeled using FITC (0.01% in DMSO, excitation/emission wavelengths = 494/518 nm) and Nile red (0.01 wt% in alcohol, excitation/emission wavelengths = 488–530/575–580 nm), respectively. A drop of sample was put on the microscope slide and then a cover slip was placed on top. The microstructure was observed using a 40× objective lens.

### 2.6. Droplet Mobility Measurements

The microrheology of the emulsions was measured using a diffusing wave spectroscopy (DWS) instrument (Rheolaser Master, Formulation Inc., Toulouse, France). The thermal motion of the colloidal particles in the samples was tracked to determine the mean square displacement (MSD). The viscoelastic properties of the samples are reflected in the relationship between the MSD and decorrelation time. All the measurements were carried out at 25 °C for 2 h.

### 2.7. In Vitro Gastric Digestion

In order to study the resistance of the emulsions to aggregation under gastric conditions, the gastrointestinal fate of the emulsions was measured using an in vitro simulated digestion model based on the INFOGEST protocol [22]. Simulated gastric fluid (SGF) was prepared in advance. The sample and SGF were preheated to 37 °C and then mixed at a ratio of 1:2 (*v*/*v*). The sample was then adjusted to pH 3.0 using 3.0 or 0.25 M HCl solutions. Subsequently, pepsin solution (2000 U/mL) was added to the mixture under continuous stirring at 100 rpm. The digestion experiment was carried out for 2 h in a water bath (37 °C). The samples were collected at regular time intervals for characterization. It should be noted that some ions were present in the original SGF solution: Na^+^, K^+^ and Ca^2+^ concentrations were 72.2, 7.8 and 0.15 mM, respectively.

### 2.8. Preparation and Characterization of Salad Dressings

Salad dressings were prepared according to the method described by Diftis et al. [23]. 1 g WPI dispersed in 99 g citrate buffer (100 mM, pH 3.8) containing 300 mM Na^+^ and 30 mM Ca^2+^ by a magnetic mixer to obtain 1 wt% WPI solution as the aqueous phase, and then it was stored at 4 °C overnight to ensure hydration. The model salad dressing was prepared by adding 50 g oil phase (100% MCT or 70% MCT + 30% CA) into 50 g aqueous phase and then homogenized for 1 min at 20,500 rpm using high-speed disperser. Salad dressings contained 50.0% MCT or 35% MCT + 15% CA, 5.0% citric acid, 2.5% NaCl, 1.5% CaCl_2_.

Inverted fluorescence microscopy (ECLIPSE Ti-s, NIKON, Tokyo, Japan) was used to observe the microstructures of salad dressing. The protein was labeled with FITC and the oil phase was labeled with Nile red. The two salad dressings were stained with Nile red and centrifuged at 2000 rpm for 10 min to observe the situation of oil separation from the system. The glass bottles containing salad dressing were tilted at a certain angle and then returned to the upright state to observe the wall-cling properties of the salad dressing.

The viscosity of salad dressing was performed on a rheometer (AR2000ex, TA, New Castle, DE, USA) equipped with 60 mm parallel plate. The test was carried out at 25 °C and waited for 60 s of thermal equilibrium. The flow sweep test was first conducted with increasing shear rate from 0.01 s^−1^ to 100 s^−1^, and then decreasing from 100 s^−1^ to 0.01 s^−1^. The whole process was carried out twice in succession. The change of the viscosity as a function of shear rate was recorded.

### 2.9. Statistical Analysis

All experiments were performed in triplicate on freshly prepared samples. The results were then reported as averages and standard deviations of these measurements. Statistical analysis was performed by one-way ANOVA using IBM SPSS Statistics 22. Samples were considered to be statistically significant if *p* < 0.05.

## 3. Results and Discussion

### 3.1. Influence of Ion Type and Strength on the Physical Stability of Emulsions

In the present work, ions with different dimensions and valences were selected to study the influence of salt on the physical stability of the protein-coated oil droplets: Na^+^, K^+^, Ca^2+^, Mg^2+^ and Fe^2+^. Preliminary experiments indicated that the effect of the different ions on emulsion stability was different, and so the concentration ranges tested were based on ion type.Figure 1 shows the appearance of WPI-stabilized emulsions in the absence (M100C0) and presence (M70C30) of cinnamaldehyde at different ion concentrations. After preparation, all the fresh emulsions were visually stable and homogenous, except after the addition of Fe^2+^ ions. After 7 days’ storage, the emulsions containing monovalent ions still appeared relatively homogenous, but those containing divalent ions exhibited visible phase separation. Even so, thin cream layers were observed at the top of the M100C0 emulsions when the concentration of monovalent ions (Na^+^, K^+^) exceeded 400 mM, which suggested that some flocculation and creaming had occurred. On the other hand, all the M70C30 emulsions appeared to be resistant to phase separation, which suggests that the incorporation of the cinnamaldehyde inhibited droplet aggregation. In the presence of divalent ions (Ca^2+^, Mg^2+^ and Fe^2+^), phase separation was observed in the M100C0 emulsions at and above a critical concentration: 10 mM for Ca^2+^, 20 mM for Mg^2+^ and 4 mM for Fe^2+^ (Figure 1). Conversely, the M30C30 emulsions had much better tolerance to phase separation in the presence of the multivalent ions, which again indicates that the cinnamaldehyde was effective at inhibiting droplet aggregation. The results indicated that divalent ions were more effective at inducing instability in the emulsions (especially Fe^2+^), which can be attributed to ion binding, ion bridging, and electrostatic screening effects. Notably, the M70C30 emulsions had a yellowish color, which was attributed to a Schiff-base reaction between the whey proteins and cinnamaldehyde [21].

The light scattering measurements indicated that the presence of divalent ions led to larger particle sizes than the presence of monovalent ions (Figure 2a–e), which is consistent with the creaming stability analysis. In the presence of monovalent ions, the mean particle diameter (Z-average) of the M100C0 emulsions first appeared to increase and then to decrease as the ion concentration was raised from 0 to 2000 mM. This trend was even more obvious after 7 days’ storage. The decrease in particle size observed at high salt concentration was probably an artefact of the dynamic light scattering (DLS) method used to analyze these emulsions. DLS measures the size of particles from an analysis of their Brownian motion in solution. As a result, it is only sensitive to colloidal particles with diameters between about 3 and 3000 nm. Any larger particles move so slowly they cannot be accurately detected. It is therefore likely that a high fraction of the protein-coated oil droplets had strongly aggregated and formed large flocs in the M100C0 emulsions containing high concentrations of monovalent ions, and so they could not be detected. Instead, the instrument only measured the size of any smaller droplets within the emulsions. Nevertheless, the DLS measurements did indicate that extensive droplet aggregation had occurred after 7 days’ storage in the M100C0 emulsions at 200 mM monovalent ions. Interestingly, the particle size was larger in the emulsions containing K^+^ than in the ones containing Na^+^ after aggregation was first observed, which suggests that the potassium ions may have been more effective at promoting irreversible flocculation of the WPI-coated oil droplets. The ability of the monovalent ions to promote droplet aggregation can mainly be attributed to their ability to screen the electrostatic repulsion between them [24].

In contrast, the M70C30 emulsions maintained a relatively small particle size (150–200 nm) at all monovalent ion concentrations (0 to 2000 mM) after 7 days’ storage, which suggests that the presence of the cinnamaldehyde increased the salt stability of these emulsions, which is consistent with previous studies [21]. In contrast, the Na^+^ ion tolerance of emulsions formulated with WPI-xanthan gum or casein-chitosan complexes was only 100 mM [25,26]. The cinnamaldehyde may increase the salt stability of the emulsions by causing the protein molecules to undergo conformational changes at the oil–water interface, which reduce the number of nonpolar groups exposed to water. As a result, the hydrophobic attraction between the protein-coated oil droplets is increased, so that more salt must be added to overcome the electrostatic and steric repulsive forces. Moreover, the presence of the cinnamaldehyde may have promoted chemical crosslinking of nonadsorbed proteins to the protein-coated oil droplet surfaces, thereby increasing the thickness of the interfacial layer [27,28]. As a result, the magnitude and range of the steric repulsion between the droplets increased, which inhibited droplet aggregation by preventing them from coming close together [29].

Figure 2a’,b’ show that the zeta-potential of the emulsions decreased slightly with increasing salt concentration. Similarly, Appendix A shows that the zeta-potential of WPI solutions also decreases with increasing NaCl and CaCl_2_ concentration. These results can be attributed to electrostatic screening and ion binding effects, which reduce the surface potential at the oil droplet surfaces [28]. The confocal fluorescence microscopy images (Figure 3) largely supported the light scattering results. For the monovalent ions, extensive droplet aggregation was observed at 200 mM and higher for the M100C0 emulsions but not droplet aggregation was observed from 200–2000 mM for the M70C30 emulsions. For the Ca^2+^, Mg^2+^ and Fe^2+^ ions, extensive droplet aggregation was observed at 10, 10 and 3 mM and above for the M100C0 emulsions but at 10, 40 and 4 mM for the M70C30 emulsions, respectively. Thus, these results show that the cinnamaldehyde was effective at increasing the salt stability of the protein-coated droplets, especially for the monovalent ions.

Interestingly, the minimum amounts of multivalent ions required to promote droplet aggregation depended on ion type. An appreciably lower concentration of Fe^2+^ ions was required than Ca^2+^ or Mg^2+^ ions. In the presence of divalent ions, the zeta-potential of the emulsions decreased more significantly as their concentration increased than for the monovalent ions (Figure 2c’–e’). This effect can be attributed to ion binding effects, as well as to electrostatic screening effects that increase as the valence of the ions increases [29,30]. For the effect of salt on the formation or property of emulsions, some research has also indicated that the presence of some salt in the emulsifiers or stabilizers might benefit to form emulsions with smaller particle size or better stability [10,11]. Since the salt affected the structure or wettability of emulsifiers or stabilizers, their interfacial absorption behavior was altered. This is different from our work, which assessed the effect of salt on the emulsion droplets.

### 3.2. Impact of Ion Type and Concentration on Droplet Mobility

The aggregation of the oil droplets in an oil-in-water emulsion should lead to a reduction in their movement due to Brownian motion, which can be monitored by measuring changes in their diffusion coefficient. For this reason, we used a microrheometer based on dynamic light scattering measurements to characterize changes in droplet diffusion in the presence of different types and concentrations of mineral ions. Specifically, the mean square displacement (MSD) versus decorrelation time of the droplets in the emulsions was measured (Figure 4). For the monovalent salts, the MSD values increased with increasing decorrelation time, which suggests that these samples were predominantly viscous (rather than elastic). At a fixed decorrelation time (0.01 s), there was no change in the MSD, independent on salt concentration (Figure 5). Conversely, for some of the divalent salts, the MSD values first increased, then reached a plateau, and then increased again with increasing decorrelation time, exhibiting an S-shaped profile, which suggests that they were predominantly elastic. This behavior is consistent with the formation of a 3D network of aggregated oil droplets that extends throughout the entire volume of the emulsion so that the individual droplets cannot move freely due to the constraints of the network [31]. At a fixed decorrelation time (0.01 s), there was a pronounced decrease in the MSD when a certain salt concentration was exceeded (Figure 5). This effect can be attributed to a reduction in the mobility of the oil droplets when they aggregated with each other and formed large flocs. As mentioned earlier, the aggregation of the oil droplets can be attributed to electrostatic screening effects for the monovalent ions, and electrostatic screening, ion binding, and ion bridging effects for the divalent ones.

The presence of cinnamaldehyde in the emulsions influenced the ion concentration where there was a large reduction in the MSD, i.e., where extensive droplet flocculation occurred. For instance, this critical concentration was around 10 mM for Ca^2+^, 20 mM for Mg^2+^ and 3 mM for Fe^2+^ in the M100C0 emulsions, and 10 mM for Ca^2+^, 40 mM for Mg^2+^ and for 4 mM Fe^2+^ in the M70C30 emulsions. Thus, the addition of cinnamaldehyde to the oil phase appeared to enhance the salt-stability of the emulsions. These results are therefore consistent with the particle size (Figure 2) and confocal microscopy (Figure 3) results reported earlier, as well as with previous studies [32].

### 3.3. Gastric Digestion Behavior of Emulsions in the Presence of Ions

Protein-coated oil droplets are known to have poor resistance to aggregation in the gastric fluids within the stomach due to the highly acidic conditions, high ionic strength, and enzyme activity there [33]. Our previous study showed that incorporating cinnamaldehyde into protein-coated oil droplets improved their resistance to aggregation in the stomach phase [32]. Here, we characterized the impact of cinnamaldehyde on the stability of protein-coated oil droplets to aggregation in the stomach when there were different types and concentrations of mineral ions present.

Based on our previous results and the actual levels of mineral ions found in food, the following ion concentrations were used: 100, 200, 80 and 1 mM for Na^+^, K^+^, Ca^2+^ and Fe^2+^, respectively. Figure 6 shows the change of pH during gastric digestion. In all emulsions, the pH increased slightly during the first 15 min and then remained fairly steady. For the monovalent ions, the overall change in pH was slightly lower in the presence of cinnamaldehyde but there was little difference for the divalent ions.

All the emulsions underwent extensive gravitational separation (creaming or sedimentation) after digestion (Appendix A). In the absence of cinnamaldehyde, all of the emulsions exhibited creaming, with a white layer of oil droplets on top of the samples. In the presence of cinnamaldehyde, creaming also occurred in the emulsions containing Na^+^, K^+^ and Mg^2+^ but surprisingly sedimentation was observed in the emulsions containing Fe^2+^. This suggests that the oil droplet-protein-mineral complexes formed in this latter case were denser than water. The thickness of the cream or sediment layer formed in the emulsions was thicker in the presence of cinnamaldehyde than in its absence. This may have been because the cinnamaldehyde increased the strength of the protein–protein interactions, which led to the formation of a more open and porous 3D network of aggregated oil droplets.

In the absence of cinnamaldehyde, there was a pronounced increase in mean particle diameter in all the WPI emulsions with increasing digestion time (Figure 7). However, in the presence of cinnamaldehyde, the increase in mean particle size was more modest in most of the emulsions, which suggests that it was able to increase the resistance of the droplets to aggregation during gastric digestion. It should be noted, however, that an increase in particle size was still observed in the presence of Na^+^.

The confocal microscopy images also showed that aggregates were formed in the WPI emulsions during gastric digestion, whose size depended on ion type and cinnamaldehyde addition (Figure 8). Particularly large aggregates were observed in the M100C0 emulsions containing Ca^2+^ at longer digestion times, which is consistent with the particle size measurements (Figure 7). The aggregates formed in the M70C30 emulsions were considerably smaller than those formed in the M100C0 ones, regardless of ion type, which again showed that the cinnamaldehyde was able to inhibit droplet aggregation. Droplet aggregation may be caused by the reduction in electrostatic repulsion, increase in bridging effects, and hydrolysis of adsorbed proteins in the gastric environment [34,35,36].

### 3.4. Application in Salad Dressings

Figure 9 presents the appearance, microstructure and viscosity of the salad dressings. Both of the salad dressings were homogenous and the one in the presence of CA had a light-yellowish color. After one day’s storage, creaming was observed in the samples, and M70C30 salad dressing showed higher creaming layer, indicating higher stability against gravitational separation. The microscopy images (Figure 9A) indicated that M70C30 salad dressing had smaller particle size. During storage, we found the free oil and oil coalescence in the M100C0 salad dressing, while M70C30 kept stable. In addition, M70C30 salad dressing had a strong wall hanging phenomenon, while M100C0 salad dressing was mild (Figure 9B), which was related to the differences of their viscosity. According to Figure 9C, within the set shear rate range, both salad dressings distinctly exhibited shear-thinning behaviors, which were identified as non-Newtonian fluids [37]. This was the ideal flow behavior characteristic of O/W emulsion salad dressing [38]. At the same time, we also found that they had thixotropic behavior, and the collapsed internal structure could be reconstructed at low shear rate. Significantly, the viscosity of M70C30 was two orders of magnitude higher than that of M100C0. The viscosity of M100C0 was derived from the cross-linking of Ca^2+^ between adjacent protein molecules, while there was also network structure formed by the reaction of protein and CA in M70C30 and therefore it had a high viscosity. The results of rheological tests also confirmed the wall hanging phenomenon.

## 4. Conclusions

In conclusion, the salt tolerance of WPI-stabilized emulsions was investigated in the absence and presence of cinnamaldehyde. As expected, divalent ions were more effective at promoting droplet aggregation than monovalent ones, which can be attributed to ion binding and ion bridging effects. As a result, the minimum amounts of mineral ions required to cause extensive aggregation of the protein-coated droplets depended on ion type, being much lower for divalent ions than monovalent ones. The addition of cinnamaldehyde to the oil phase of the emulsions was highly effective at inhibiting salt-induced droplet aggregation. For instance, in the case of monovalent ions, droplet aggregation could be suppressed over a very wide range of ion concentrations (0 to 2000 mM). This phenomenon may be useful when designing emulsions that need to remain stable in certain high-salt foods, such as some sauces and dressings. The presence of cinnamaldehyde was also shown to increase the resistance of the protein-coated oil droplets to aggregation under simulated gastric conditions. Some research suggests that gastric-stable emulsions may be able to increase the satiety response, which could reduce the tendency to overeat. Overall, the knowledge gained in this study might contribute to the development of salt- and gastric-resistant emulsions that might be beneficial for applications in high-salt sauces and diet foods.

## Figures and Tables

**Figure 1 foods-10-01868-f001:**
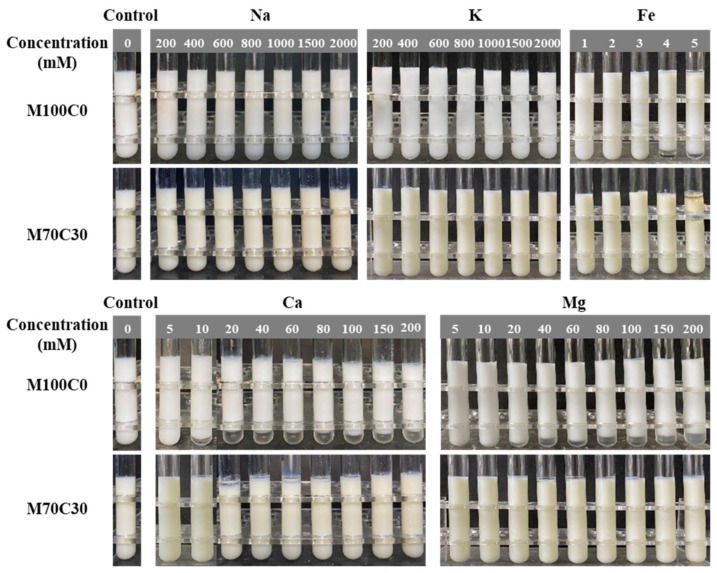
Appearance of WPI-stabilized emulsions in the absence and presence of CA at different ion types and strengths after 7 days’ storage. M100C0 represents that the oil phase is 100 wt% MCT. M70C30 represents that the oil phase is a mixture of 70 wt% MCT and 30 wt% CA.

**Figure 2 foods-10-01868-f002:**
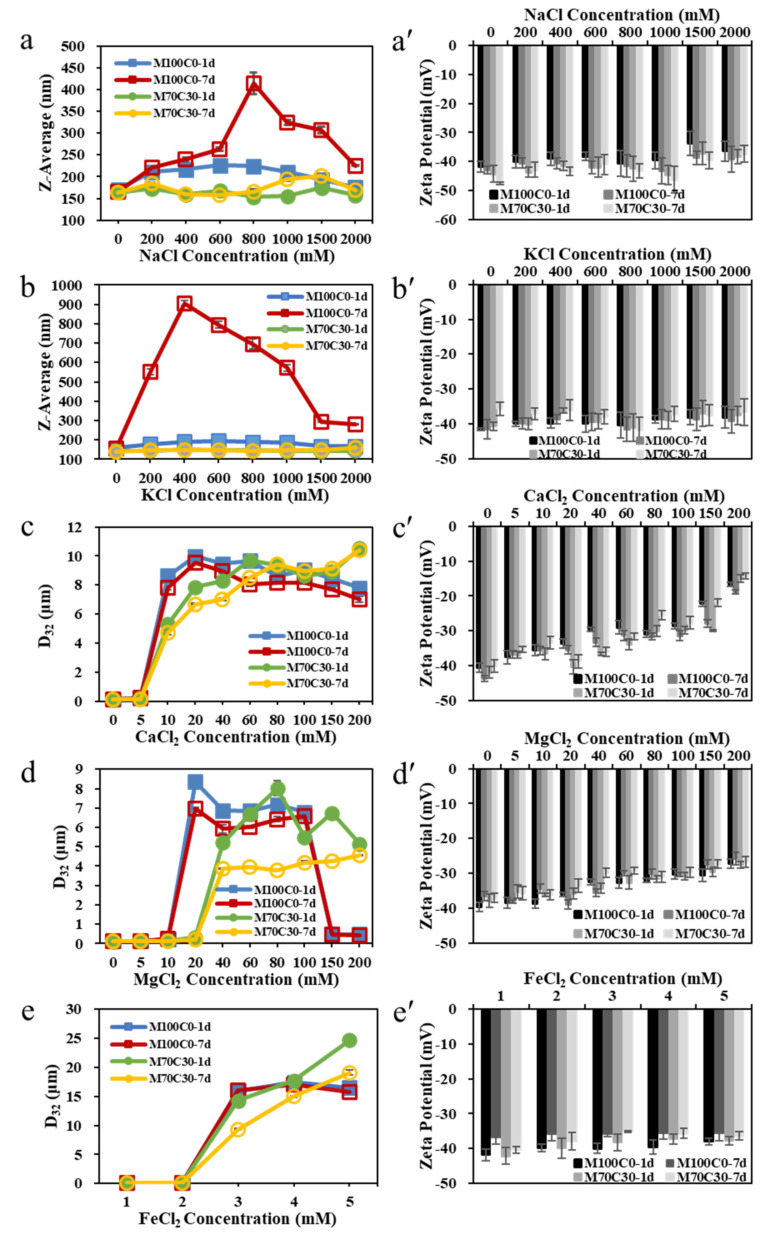
Particle size (**a**–**e**) and ζ-potential (**a’**–**e’**) of WPI-stabilized emulsions at different ion types and strengths after 1 day’s and 7 days’ storage.

**Figure 3 foods-10-01868-f003:**
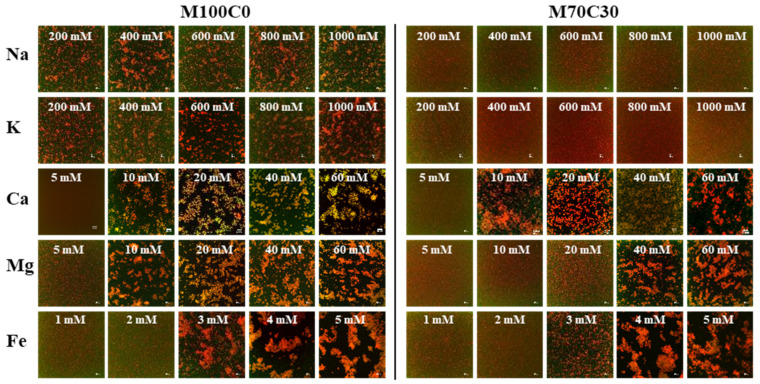
CLSM images of WPI-stabilized emulsions at different ion types and strengths. The pictures are overlays of Nile red-labeled oil phase and FITC-labeled protein. The scale bar is 20 μm.

**Figure 4 foods-10-01868-f004:**
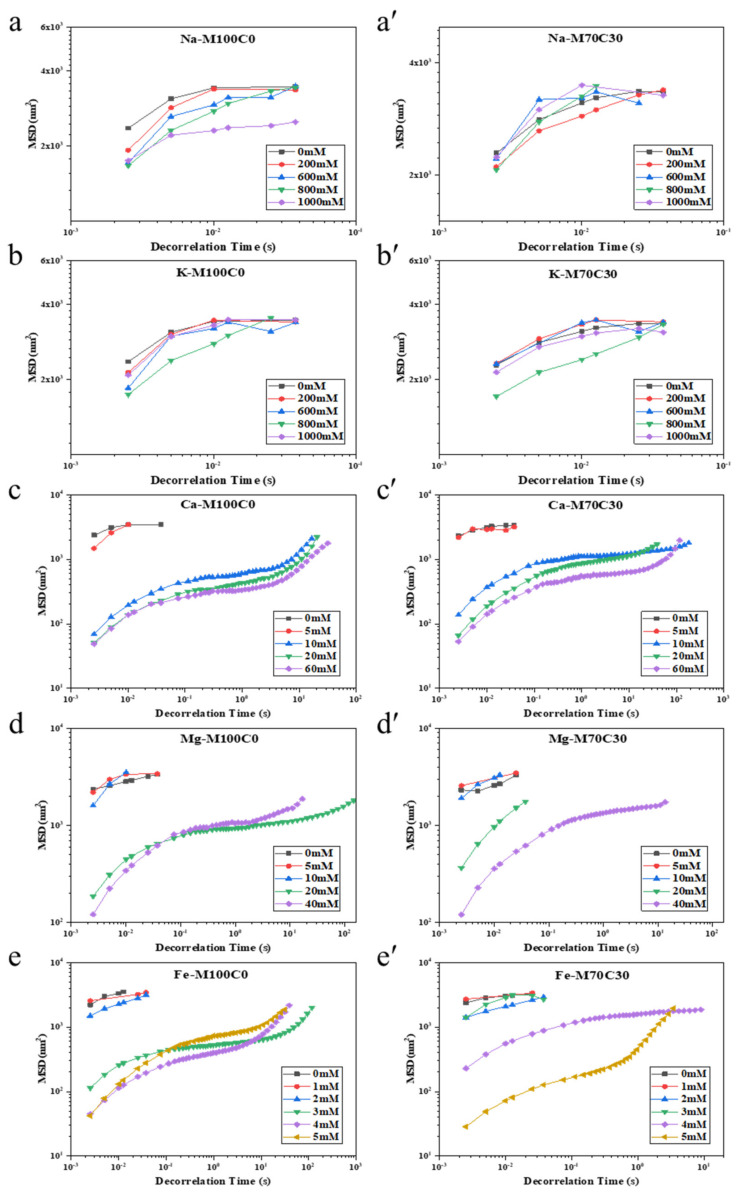
MSD profile versus decorrelation time curves of WPI stabilized emulsions in the absence (**a**–**e**) and presence (**a’**–**e’**) of CA as a function of ionic strengths.

**Figure 5 foods-10-01868-f005:**
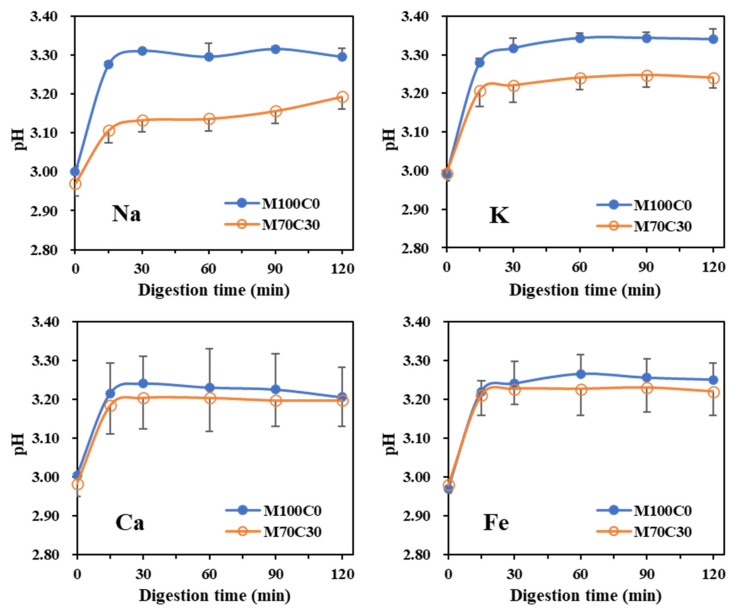
Changes of pH values for emulsions during digestion.

**Figure 6 foods-10-01868-f006:**
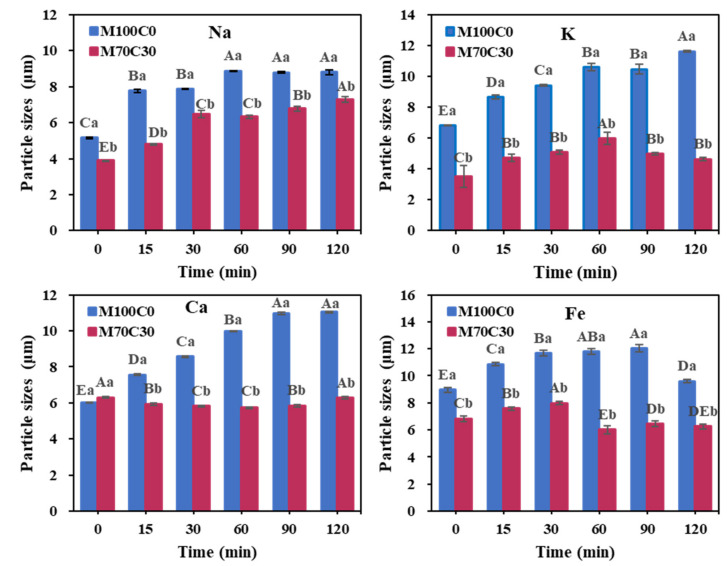
Particle sizes of the two emulsions with different ions after digestion. Different capital letters represent significant differences in the adsorption rate of the same emulsion with different ionic strengths. Different lowercase letters represent significant differences in the adsorption rate of different kinds of emulsion with the same ionic strength. *p* < 0.05.

**Figure 7 foods-10-01868-f007:**
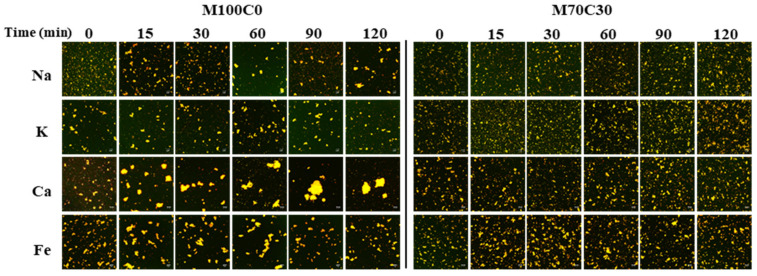
CLSM images of emulsions with different ions after digestion. The pictures are overlays of Nile-red-labeled oil phase and FITC-labeled protein. The scale bar is 20 μm.

**Figure 8 foods-10-01868-f008:**
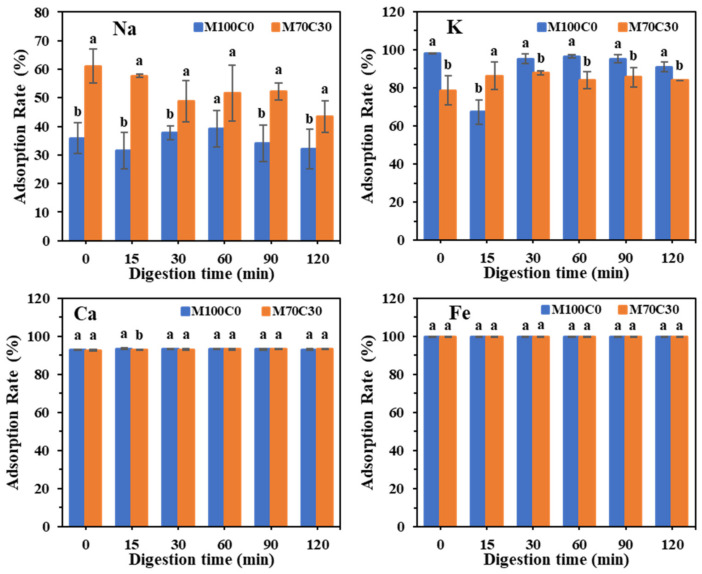
Adsorption rates of different ions on protein after digestion. Different lowercase letters indicate a significant difference in ion adsorption rates between the two emulsions after the same digestion time. *p* < 0.05.

**Figure 9 foods-10-01868-f009:**
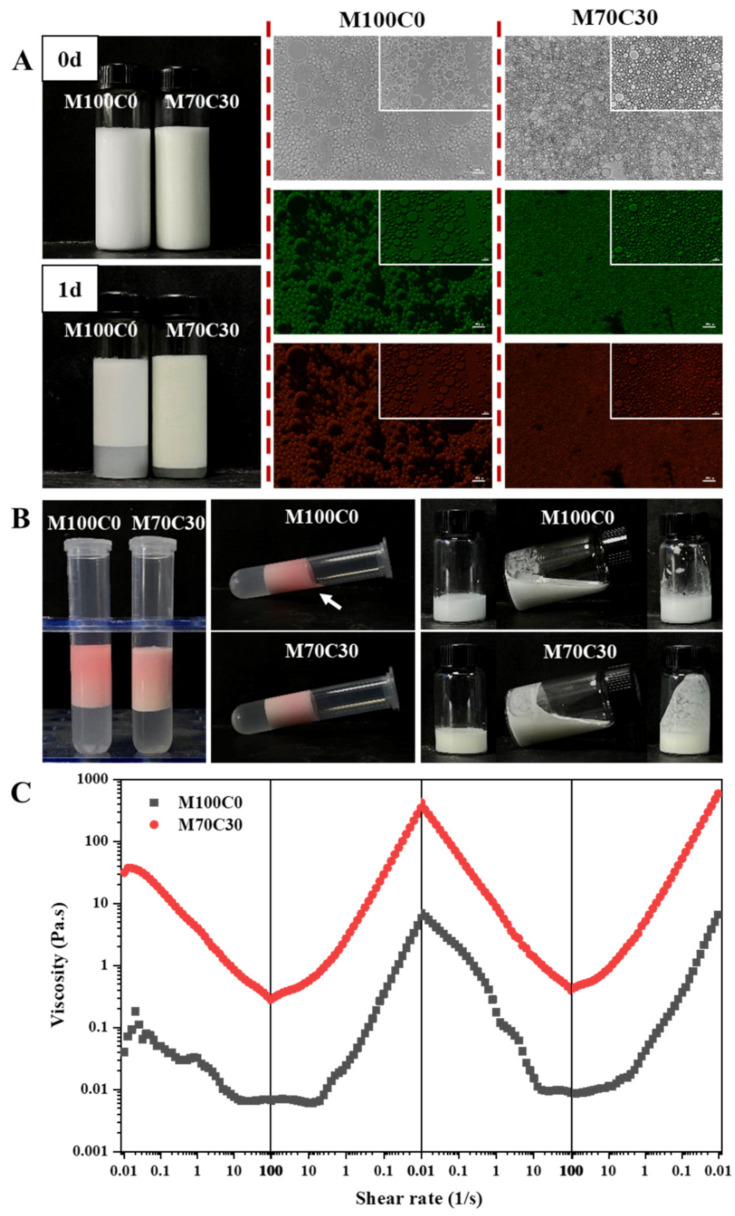
(**A**) Appearance and microstructure of model salad dressing added Na+ and Ca2+ in the presence and absence of CA. The green pictures are FITC-labeled protein and the red pictures are Nile-red-labeled oil phase. The scale bar is 20 μm. (**B**) The images of model salad dressing centrifugation after dyeing and the wall-cling property. (**C**) Viscosity versus shear rate curves of model salad dressing added Na^+^ and Ca^2+^ in the presence and absence of CA.

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
