# Peer review of "Development of Salt- and Gastric-Resistant Whey Protein Isolate Stabilized Emulsions in the Presence of Cinnamaldehyde and Application in Salad Dressing"

_foods, 2021, doi:10.3390/foods10081868_

Round 1
Reviewer 1 Report
Dear Editor and Authors,
I send you my review about the paper “Development of salt- and gastric-resistant whey protein isolate-stabilized emulsions in the presence of cinnamaldehyde and application in salad dressing”.
The purpose of the paper, as reported in the title was to develop of stabilized emulsions based on milk whey protein and resisting to salt and gastric digestion.
The paper result well written in a good English and well structured.
However, in this form it show some lacks and need some revision that I reported below.
The introduction, result well written and adequately supports the aim of the paper.
However, at the end of the introduction the sentences that goes from “Monovalent (at line 70) to “was evaluated” (at line 74) should be remove and shift in Materials and methods paragraph.
To this aim I suggest to the Authors to create a specific paragraph “experimental design” to be inserted before the paragraph “2.1 Materials”.
Moreover, in the section “Materials” the whey protein isolate composition is probably wrong.
Indeed, the sum of the 93% of protein, 1% of fat, 4,5% of moisture and 2% of ash gives as a result 100.5%.
Always in the “Materials and method” paragraph in the section “2.9 Statistical analysis” the method used for used to calculate the significance of the difference should be reported. For example Analysis of variance o T of student. Moreover, it should be reported, also, if were calculated the least square means, and, in this case it should be reported the model used.
The results is well presented and adequately discussed, also, in relation to the references reported that it is complete.
Finally, the conclusions are in accordance with the data discussed and with the aim of the document, however, at the line 397 instead of writing “some applications” should be better to report 2 or 3 application as example.
Best regards
Author Response
No 1 Dear Editor and Authors,
I send you my review about the paper “Development of salt- and gastric-resistant whey protein isolate-stabilized emulsions in the presence of cinnamaldehyde and application in salad dressing”.
The purpose of the paper, as reported in the title was to develop of stabilized emulsions based on milk whey protein and resisting to salt and gastric digestion.
The paper result well written in a good English and well structured.
However, in this form it show some lacks and need some revision that I reported below.
The introduction, result well written and adequately supports the aim of the paper.
However, at the end of the introduction the sentences that goes from “Monovalent (at line 70) to “was evaluated” (at line 74) should be remove and shift in Materials and methods paragraph. To this aim I suggest to the Authors to create a specific paragraph “experimental design” to be inserted before the paragraph “2.1 Materials”.
Response: we have revised the paper based on the comments.
Experimental design: ‘The stability of the emulsions against ionic strength was first studied by characterizing particle size, zeta potential, microstructure and the droplet mobility. Then, the resistance of emulsions to aggregation under gastric digestion conditions was investigated. Finally, the emulsion-based salad dressing product was developed.’
Moreover, in the section “Materials” the whey protein isolate composition is probably wrong.
Indeed, the sum of the 93% of protein, 1% of fat, 4,5% of moisture and 2% of ash gives as a result 100.5%.
Response: The composition was provided by the commercial product. We modified the protein content as the dry basis. Protein content was 89% as it is.
“which consisted of 93% protein (dry basis), 1.0% fat, 4.5% moisture, and 2.0% ash.”
Always in the “Materials and method” paragraph in the section “2.9 Statistical analysis” the method used for used to calculate the significance of the difference should be reported. For example Analysis of variance o T of student. Moreover, it should be reported, also, if were calculated the least square means, and, in this case it should be reported the model used.
Response: We have added the significance analysis method in the section 2.9 Statistical analysis. “Statistical analysis was performed by one-way ANOVA using IBM SPSS Statistics 22.”
The results is well presented and adequately discussed, also, in relation to the references reported that it is complete.
Finally, the conclusions are in accordance with the data discussed and with the aim of the document, however, at the line 397 instead of writing “some applications” should be better to report 2 or 3 application as example.
Response: We modified some applications into "high salt sauces and diet foods".
Reviewer 2 Report
The work is very interesting and extensive. Here are some small tips to improve your work.
I agree with authors that the role of salt in oil-in-water emulsion is not widely studied, but relevant to food processing, formulation and sensory properties. The literature review has been well compiled, but the authors should mention a recent similar work (Doğan et al. 2020) in which this topic is addressed. For research purposes, please describe what new this work brings. In another recent work Influence of salt concentration on the formation of Pickering emulsions (French et al. 2020), was studies, please include this in the introduction. Discussion section should be improved by adding proposed literature.
Methods
Please explain why such concentrations of salt ions have been used where such concentrations may exist. Please provide examples
Figure 6 – above the columns, please provide letters that indicate significant differences
Figure 8. in the figure on adsorption rates of Ca ions there is b letter in the figure on Adsorption rates of Ca ions there is b letter above the 3rd column. are you sure if it shouldn't be there? and it doesn't seem different from others
Doğan, M., Göksel Saraç, M., & Aslan Türker, D. (2020). Effect of salt on the inter-relationship between the morphological, emulsifying and interfacial rheological properties of O/W emulsions at oil/water interface. Journal of Food Engineering, 275. https://doi.org/10.1016/j.jfoodeng.2019.109871
French, D. J., Fowler, J., Taylor, P., & Clegg, P. S. (2020). Influence of salt concentration on the formation of Pickering emulsions. Soft Matter, 16(31), 7342–7349. https://doi.org/10.1039/d0sm00321b
Author Response
The work is very interesting and extensive. Here are some small tips to improve your work.
I agree with authors that the role of salt in oil-in-water emulsion is not widely studied, but relevant to food processing, formulation and sensory properties. The literature review has been well compiled, but the authors should mention a recent similar work (Doğan et al. 2020) in which this topic is addressed. For research purposes, please describe what new this work brings. In another recent work Influence of salt concentration on the formation of Pickering emulsions (French et al. 2020), was studies, please include this in the introduction. Discussion section should be improved by adding proposed literature.
Response: As suggested by reviewer, we added those literatures in the introduction. The literatures provide useful information. In their work, they studied the effect of salt on the emulsifiers and particles, and then the further effect on the formation of emulsions. In the present work, we prepared the emulsions first and then investigated the stability of emulsion in the salt solutions.
In the introduction: ‘A recent work by Doğan et al confirmed the effect of salt (NaCl and CaCl2) on emulsions and also provided the effect mechanism of salt by affecting the interfacial rheological property of emulsions [10]. They demonstrated that the interfacial rheology results could predict the emulsion stability and CaCl2 salt increased the stability of emulsions stabi-lized by the selected emulsifiers (lecithin, mono-diglycerides, sodium steoryl-2-lactylate), as compared to NaCl salt. Moreover, Pickering emulsions have been known to have better stability against coalescence and aggregation. French et al found that salt affected the par-ticle-particle interaction, wettability of particle and also the particle-interface interaction. [11]. They concluded when considering the formation of Pickering emulsions, the influ-ence of salt on the particle–particle interaction was less important than the influence of salt on both the particle wettability and the particle–interface interaction.’
In the results: ‘For the effect of salt on the formation or property of emulsions, some work also indicated that the presence of some salt in the emulsifiers or stabilizers might benefit to form emul-sions with smaller particle size or better stability [10, 11]. Since the salt affected the struc-ture or wettability of emulsifiers or stabilizers, their interfacial absorption behavior was altered. This is different from our work, which assessed the effect of salt on the emulsion droplets.’
Methods
Please explain why such concentrations of salt ions have been used where such concentrations may exist. Please provide examples
Response: we added some applications in the introduction, like sauces.
Figure 6 – above the columns, please provide letters that indicate significant differences
Response: We have added significance analysis to each column and modified the corresponding chart description.
Figure 8. in the figure on adsorption rates of Ca ions there is b letter in the figure on Adsorption rates of Ca ions there is b letter above the 3rd column. are you sure if it shouldn't be there? and it doesn't seem different from others
Response: After checking the data, we determined that it was correct, and P value was 0.046, calculated by one-way ANOVA analysis. The error bar is smaller than the others.
Reviewer 3 Report
Title:
It seems adequate within the context of the investigation
Abstract:
The abstract is largely descriptive. I would suggest adding more numerical results within the description.
Introduction:
No comments
Materials & Methods:
- Line 94: it is not clear why the mass ratio 7:3 was selected
- Line 102: What was the volume?
Author Response
Title:
It seems adequate within the context of the investigation
Abstract:
The abstract is largely descriptive. I would suggest adding more numerical results within the description.
Response: We have added numerical results in the abstract.
Introduction:
No comments
Materials & Methods:
- Line 94: it is not clear why the mass ratio 7:3 was selected
Response: Previously, we have reported WPI-stabilized emulsions was modified by CA (Chen, et al, 2018). Hence, the ratio of MCT and CA was selected based on our previous report.
- Line 102: What was the volume?
Response: In our opinion, reviewer might want us to mention the volume of emulsions during preparation. When preparing emulsions, we tried our best to avoid the dilution of water through the homogenization. Usually, the mass of the formulation was 100 g. After homogenization, the volume of fine emulsion was around 85-95 g. To be honest, we did not really measure the volume of emulsion after preparation.